# Genetic Polymorphisms of Vascular Endothelial Growth Factor in Neonatal Pathologies: A Systematic Search and Narrative Synthesis of the Literature

**DOI:** 10.3390/children10040744

**Published:** 2023-04-19

**Authors:** Monica G. Hăşmăşanu, Lucia M. Procopciuc, Melinda Matyas, Gabriela I. Zonda, Gabriela C. Zaharie

**Affiliations:** 1Department of Neonatology, Iuliu Haţieganu University of Medicine and Pharmacy, 400006 Cluj-Napoca, Romania; popa.monica@elearn.umfcluj.ro (M.G.H.);; 2Department of Medical Biochemistry, Iuliu Haţieganu University of Medicine and Pharmacy, 400349 Cluj-Napoca, Romania; 3Department of Mother and Child Care, “Grigore T. Popa” University of Medicine and Pharmacy Iasi, 700115 Iași, Romania

**Keywords:** vascular endothelial growth factor (VEGF), polymorphism, neonatal diseases

## Abstract

(1) Background: Vascular endothelial growth factor (VEGF) is essential in vasculo- and angiogenesis due to its role in endothelial cell proliferation and migration. As a vascular proliferative factor, VEGF is one of the hallmarks of cancer and, in adult populations, the relationship between genetic polymorphism and neoplasm was widely investigated. For the neonatal population, only a few studies attempted to uncover the link between the genetic polymorphism of VEGF and neonatal pathology, especially related to late-onset complications. Our objective is to evaluate the literature surrounding VEGF genetic polymorphisms and the morbidity of the neonatal period. (2) Methods: A systematic search was initially conducted in December 2022. The PubMed platform was used to explore MEDLINE (1946 to 2022) and PubMed Central (2000 to 2022) by applying the search string ((VEGF polymorphism*) and newborn*). (3) Results: The PubMed search yielded 62 documents. A narrative synthesis of the findings was undertaken considering our predetermined subheadings (infants with low birth weight or preterm birth, heart pathologies, lung diseases, eye conditions, cerebral pathologies, and digestive pathologies). (4) Conclusion: The VEGF polymorphisms seem to be associated with neonatal pathology. The involvement of VEGF and VEGF polymorphism has been demonstrated for retinopathy of prematurity.

## 1. Introduction

Vascular endothelial growth factor (VEGF) is important in vasculogenesis and angiogenesis because of its role in the migration and proliferation of endothelial cells. It is a significant mediator of vascular permeability, which is essential in fetal development. VEGF promotes the development of new blood vessels and then enhances the vascular permeability of endothelial cells.

Genetic polymorphism of VEGF was studied in the adult population, demonstrating its essential role in vascular proliferation and the appearance of neoplasms [1,2,3]. The VEGF gene is situated on chromosome 6p21. 3 and consists of 8 exons and 7 introns [1]. It is highly polymorphic, with at least 30 functional single-nucleotide polymorphisms (SNPs) in the 5′-untranslated region (UTR), 3′-UTR, and promoter regions [1]. The genetic polymorphism of VEGF has been studied in the adult population but also the neonatal population.

The polymorphism of the VEGF +1612G>A gene was found to be possibly associated with gastric cancer in a study of Japanese and Chinese Han adults [4,5]. Kim et al. [6] reported that in the Korean population that VEGF gene 936 + C>T polymorphism could be an independent predictive marker in patients with gastric cancer. Studies evaluating the relationship between VEGF polymorphism and gastric and intestinal cancer have yielded inconclusive results [7,8,9]. Feng et al. [10] showed, in a meta-analysis, that the VEGF +936C>T gene polymorphism is significantly associated with an increased risk of developing malignant digestive system tumors. Chen et al. [11] reported a modest significance of VEGF -2578C/A polymorphism as a risk factor for bladder cancer. No significant associations of VEGF +936C/T, -460C/T, and -2578C/A gene polymorphisms with lung cancer have been reported by Yang et al. [12] in a meta-analysis conducted on 13 independent case–control studies, with a total of 4477 patients with lung cancer and 4346 healthy controls.

The association of VEGF polymorphism with diabetes in adult populations has received attention from researchers. Han et al. [13] showed that +936C/T (rs3025039) is likely to be associated with susceptibility to diabetic retinopathy in Asian populations, and the recessive model of -460T/C (rs833061) is associated with elevated diabetic retinopathy susceptibility. Gong and Sun [14] demonstrated the association of diabetic retinopathy with -460T/C polymorphism of the VEGF gene, but not with 2578C/A polymorphism. VEGF polymorphisms are associated with the development of macular edema in patients with diabetic eye disease and neovascular age-related macular degeneration.

Only a limited number of studies have analyzed the neonatal population to evaluate the link between the genetic polymorphism of VEGF and pathology, especially late-onset complications. Accordingly, the objective of this study was to present a narrative synthesis of the research literature about the evaluation of VEGF genetic polymorphisms and their link with neonatal morbidities.

## 2. Materials and Methods

A systematic search was initially conducted in December 2022. The PubMed platform was used to explore MEDLINE (1946 to 2022) and PubMed Central (2000 to 2022), EMBASE and Cochrane Reviews, by applying the search string ((VEGF polymorphism*) and newborn*), limited to human subject and English language papers. No restrictions were applied regarding the study designs. The PubMed search yielded 62 documents.

The articles were included after screening the title and abstract, followed by the review of the full-text; articles not directly related to the investigation of VEGF genetic polymorphisms in neonatal populations were excluded.

Most studies reported investigation of VEGF genetic polymorphisms in the neonatal period, with four studies completed in the fetal period [15,16,17,18]. The evaluated articles included three meta-analyses [19,20,21], six reviews [15,22,23,24,25,26], and one clinical trial [27].

A narrative synthesis of the findings was undertaken considering our predetermined subheadings (infants with low birth weight or preterm birth, heart pathologies, lung diseases, eye conditions, cerebral pathologies, and digestive pathologies).

## 3. Results

### 3.1. Pregnancy Induced Hypertension, Pre-Eclampsia, Low-Birth-Weight Infants and Preterm Birth

Pregnancy is a complex physiological process that involves numerous physiological and biochemical changes in the body to support fetal growth and development. Placental vascular development initially implies vasculogenesis, followed by extensive branching of the fetal vessels by angiogenesis, as well as the interaction of several regulatory factors, such as members of the VEGF, placental growth factor (PGF), fibroblast growth factor (FGF), WNT, and the TGF-β and BMP families [28].

One of the critical factors that play a vital role in the development and maintenance of pregnancy is VEGF. This effective angiogenic factor stimulates the growth and proliferation of blood vessels, including the placental vasculature, which is indispensable for the proper supply of nutrients and oxygen to the developing fetus. Variations in VEGF expression and polymorphisms have been associated with various pregnancy complications, such as pregnancy-induced hypertension (PIH), pre-eclampsia, preterm labor, and low birth weight. The deletion of the VEGF gene generates severe defects in placental vascularization, leading to placental insufficiency and fetal demise [29]. Murine experiments showed that ablation of the VEGF receptor is lethal to mice embryos, whereas over expression of its soluble isoform induces symptoms consistent with pre-eclampsia, such as hypertension, proteinuria, and intrauterine growth restriction (IUGR) [30,31].

Pregnancy-induced hypertension (PIH) is characterized by elevated blood pressure levels, which can lead to significant maternal and fetal morbidity and mortality. It has been postulated that deranged angiogenic processes are partly responsible for microcirculatory vasoconstriction, suggesting a link between hypertension and angiogenesis and an influence of the genetic background on the occurrence and severity of hypertension disorders in pregnancy.

Preeclampsia, a severe complication occurring after 20 weeks of gestation, affects approximately 2–8% of all pregnancies worldwide [32,33,34,35] and represents the leading cause of maternal death in the industrialized world [36]. It is characterized by high blood pressure levels, proteinuria, and organ damage, which can lead to important maternal and fetal morbidity and mortality.

In humans during early pregnancy, the VEGF gene expression has critical roles on normal physiology or physiopathology of placental precursor cells [37]. Placental VEGF expression is important, and a higher expression of placental VEGF was found in women who delivered neonates with birth weight > 4000 g [38].

Keshavarzi et al. [16] found higher placental VEGF gene mRNA expression in PE women with the-634CC genotype. However, the study showed the mRNA expression of the placental VEGF gene has been up-regulated in the placenta of women with -634CC genotype but no association between VEGF mRNA expression and the placental VEGF -1154G/A and -2549 I/D polymorphisms.

Chedraui et al. [39] analyzed VEGF gene polymorphisms in pregnant women with severe PE and healthy controls and measured levels of VEGF and other markers (nitric oxide and asymmetric dimethylarginine) in the fetoplacental unit. The authors showed that VEGF levels were significantly lower in women with severe PE compared to healthy controls, and also an association between certain VEGF gene polymorphisms and severe PE. Due to the relatively small number of samples, the authors could not identify differences with statistical significance in the distribution of single-nucleotide polymorphisms (SNPs) in PE cases vs. controls. However, they did find a significant trend toward lower VEGF levels in the umbilical vein in pregnant women with PE who had -2578CC and -1154AG genotypes.

Starting from the hypothesis that two polymorphisms in KDR (-604T/C and Val297Ile) have been associated with coronary artery disease, Andraweera et al. [40] investigated the relationship between SNPs in the KDR gene (which encodes the receptor for VEGF) and gestational hypertension, PE, and small-for-gestational-age (SGA) infants. They studied blood samples collected from pregnant women, their partners, and infants, and analyzed the DNA for SNPs in the KDR gene. The authors found that 10.9% of the women included in the study developed gestational hypertension, 8.0% developed PE, and 7.7% gave birth to SGA infants in the absence of maternal hypertensive disease. KDR-604T/C CC genotype in the neonates was associated with gestational hypertension and PE. Moreover, the association was stronger in cases of SGA infants with abnormal uterine and/or umbilical artery Doppler.

These findings suggest a possible involvement of homozygosity for the C allele of the KDR-604T/C polymorphism to disturbances of placental vascular growth. In this study, the KDR-604T/C polymorphism in the mother was not related with PE or SGA infants. Additionally, the results did not show an important association of the KDR-604T/C polymorphism with PIH in the absence of SGA, nor with SGA if the mother did not develop hypertensive disease during pregnancy. Moreover, paternal homozygosity for the variant allele of the KDR-604T/C was linked with the risk of both PE and SGA [40].

Procopciuc et al. [15] reported, in a Romanian cohort, the presence of the maternal/fetal VEGF-CT936 polymorphism on maternal PE during pregnancies that resulted in low birth-weight neonates. They investigated the relationship between a genetic variant in the VEGF-C gene and the risk and severity of PE, as well as its effect on the maternal angiogenic profile and pregnancy outcome. The authors analyzed the DNA for the VEGF-C936T genetic variant in pregnant women with PE and healthy controls, as well as from their newborns. The researchers also measured levels of various angiogenic factors in the maternal blood samples. The results showed that the T allele of the VEGF-C936T genetic variant was significantly associated with an increased risk and severity of PE. Additionally, the study found that the maternal/newborn interaction of the VEGF-C936T variant was associated with a worse maternal angiogenic profile and resulted in preterm delivery and significantly lower birth-weight neonates.

The connection between VEGF polymorphisms and preterm labor have been investigated in several studies. The causes of preterm birth are multifactorial, but there is an increased focus on the role of genetic factors, particularly variants in the VEGF gene, due to its critical role in fetal growth and development.

Atis et al. [18] conducted a study to investigate the connection between polymorphism of the VEGF +813 gene in fetuses and preterm birth. The authors assessed samples from umbilical cord blood and included in the study 31 neonates born preterm, 34 born to mothers with PE, and 58 healthy term infants and found a significant association between CC polymorphism and preterm delivery. Additionally, the presence of the 813C allele instead of the T allele was associated with a 2.8-fold increase in the probability of preterm birth. This study has some limitations, such as the relatively small sample size and the fact that it was conducted in a specific population of Turkish women, in which the T allele of the VEGF gene is very rare. Thus, there was no TT homozygosity in the preterm group, and it was very rare in the term group, which made it impossible for the authors to calculate the odds ratio of CC polymorphism for preterm delivery.

The research article by Langmia et al. [41] investigated the relationship between certain polymorphisms of VEGFA gene, plasma levels of VEGFA, and spontaneous preterm birth. The authors found a significant association between VEGFA rs2010963 and genetic susceptibility to preterm birth in a Malay population at both allelic and genotypic levels. Women with preterm delivery had a significantly higher frequency of CG and GG genotypes compared to those delivered at term. The odds of the G allele occurrence among women with preterm labor was 1.8 times higher than those with term birth.

Papazoglou et al. [42] reported in their review results that imply VEGF gene polymorphisms in the pathological disorders of pregnancy, and a statistically significant genetic alteration, as gene 936C/T VEGF polymorphism correlates with the spontaneous preterm delivery.

Poggi et al. [43] reported in a retrospective study conducted on preterm neonates (*n* = 342) with a gestational age ≤ 28 weeks that the allele frequency and genotype distribution of VEGFA polymorphisms and endothelial nitric oxide synthase (eNOS) genes may independently affect birth weight and gestational age and act as protective or risk markers for prematurity complications, such as bronchopulmonary dysplasia (BPD), retinopathy of prematurity (ROP), and intraventricular hemorrhage (IVH).

Bányász et al. [44] evaluated 128 low birth-weight neonates, suggesting that VEGF G+405C polymorphism might be associated with a higher risk of preterm birth and that VEGF C-2578A polymorphism may contribute to the development of neonatal complications, such as necrotizing enterocolitis (NEC) and acute renal failure (ARF).

In summary, current data suggest that VEGF polymorphisms are emerging as important genetic risk factors for preterm birth. These variants affect the expression and secretion of VEGF, which have a critical role in placental and fetal development and could, potentially, be a target for preventive or therapeutic interventions.

### 3.2. Heart Pathologies

Congenital cardiac defects, also known as congenital heart defects (CHD), are a group of structural abnormalities of the heart that occur during fetal development. These defects are the most common type of congenital defects, affecting approximately 0.8% to 1.2% of newborns worldwide [45,46,47].

The causes of CHD are complex and multifactorial, involving both genetic and environmental factors. One potential genetic factor that has been implicated in CHD is variations in the VEGF gene. The crucial role in the development of the cardiovascular system is VEGF by facilitating the correct heart morphogenesis during the avascular phase of the heart [48] and promoting the growth of blood vessels. The mechanisms by which VEGF polymorphisms may contribute to the development of CHD are not fully understood. One possibility is that variations in the expression and function of VEGF may lead to abnormal blood vessel growth and development in the developing heart, resulting in structural abnormalities that can lead to CHD. Another possibility is that VEGF polymorphisms may affect the response of the fetal heart to hypoxia, which is a common trigger for CHD. SNPs in the VEGF gene may affect VEGF expression and function and have been connected with a higher risk of several cardiovascular diseases, including CHD [49,50,51]. In mouse embryos, over expression, and increased function of VEGF was associated with TOF, pulmonary stenosis, and ventricular septal defects [52].

The most studied VEGF gene polymorphisms are the -1154G/A (rs1570360), 2578C/A (rs699917), situated in the promoter region of the gene, and -634G/C (rs2010963), situated in the 5′UTR region of the VEGF gene. Even though these genetic segments are not transcribed in the expression of gene, the promoter region is responsible for transcription initiation. At the same time, 5′UTR is involved in key processes such as the maintaining of mRNA stability, and interaction and folding with ribosomes [53].

Yan et al. [54] investigated the connection between several VEGF SNPs and the risk of congenital heart disease, such as Tetralogy of Fallot (TOF), in a Han Chinese population. Their results showed an increased risk for TOF in carriers of the homozygous mutant genotypes of -2578C/A. Additionally, the mutant A allele of -1154G/A was related with a significantly higher risk of TOF. For -634G/C, the GC carrier status was significantly associated with susceptibility of TOF, but not for homozygote CC (mutant form), compared to homozygote GG (wild form).

Watson et al. [55] reported the association of rs36208048 polymorphism inside the regulatory VEGF region with ventricular septal defect (VSD). Additionally, Smedts et al. [56] found that -1154 G and -2578 C alleles play a role in the genetic disposition to endocardial cushion defects. They hypothesize that during the formation of the endocardial cushion, these genetic variants contribute to an increase in VEGF expression, alongside other factors, such as hyperglycemia, hyperhomocysteinemia, or hypoxia, altering the correct cardiogenesis [57,58].

Two studies investigated the association of VEGF SNPs with the occurrence of isolated VSD in Pakistani children, a population distinctive due to the impact of consanguineous marriages, as well as religious, social, and cultural factors, on the emergence of various genetic disorders. The first study analyzed the rs699947 (c.-2578C > A) variant of the VEGF gene and found a significant statistical difference in the frequencies of the A and C alleles in children with isolated VSD and controls, suggesting that the presence of the A allele can increase the rate for heart development issues (allelic odds ratio for A versus C was 2.03, with a confidence interval (CI) of 1.41–2.92 and a *p*-value of 0.0001) [59]. However, the second study selected the rs36208048 SNP of VEGF and did not find a statistical difference in allele frequencies between cases with VSD and non-VSD children [60].

Ding et al. [61] investigated four VEGF SNPs in a case–control study on 476 patients with CHD and 557 controls in a Chinese population. They found no significant associations of either -2578 C>A, -1498 T>C, -634 G>C, and +936 C>T with the risk of TOF, VSD, or any CHD. However, C-2578T-1498G-634C+936, C-2578T-1498C-634T+936, and G-2578G-1498G-634T+936 haplotypes were correlated with an increased risk of CHD, while C-2578T-1498G-634T+936 haplotypes were associated with a considerably lower risk of CHD.

Around 1/3 of cases of CHD need surgical treatment, and most of them require the use of cardiopulmonary bypass (CPB). Post-operative mortality remains high, particularly for the more severe cardiac defects. Some studies have been able to discover several genetic variations that impact vascular response pathways and contribute to the long-term survival of pediatric patients undergoing surgery.

Given the fact that VEGFA is involved in the response to ischemia and low output in patients with cardiopulmonary bypass (CPB) and in vascular adaptation to hemodynamic alterations, Kim et al. [62] investigated the effect of VEGF polymorphism on survival in 422 non-syndromic CHD children who underwent cardiac surgery before six months of life after cardiopulmonary bypass. The results of the survival analysis revealed that the intronic SNP rs833069 in VEGFA was linked to long-term survival (HR = 0.37, *p* = 7.03 × 10^−4^). Moreover, the survival probability increased in a dose-dependent manner for each minor allele of VEGFA SNP rs833069, meaning that for the minor allele, each copy was associated with a rise in long-term transplant-free survival [62]. A follow-up study by Mavroudis et al. [63] provided evidence that the preservation of ventricular function may be the mechanism through which the minor allele of the VEGFA SNP rs833069 improves survival.

Genetic variations in the VEGFA gene have been linked to non-syndromic CHD, but also to cardiovascular anomalies in individuals with microdeletion 22q11. Patients with del22q11 frequently develop CHD, necessitating complex surgical interventions [64], and they have a higher immediate postoperative mortality rate than patients with non-syndromic CHD or other syndromes.

Calderón et al. [65] examined the association between three VEGFA polymorphisms (c.-2578C>A, c.-634C>G, and c.-1154G>A) and CHD in Chilean patients with microdeletion 22q11, in a family-based association and case–control study. They included 122 patients and their parents, half of whom had CHD. The researchers demonstrated the partial penetrance of the phenotype for microdeletion 22q11.Still they found no evidence of an association between polymorphisms of VEGFA and CHD, by either case–control or transmission disequilibrium testing analysis. The results of this study are in contrast to previous findings from a Flemish group [66] that showed a higher frequency of VEGFA promoter SNPs in patients with CHD. The researchers noted that these disparities could be due to ethnic differences or a lack of association in Chilean patients.

Studies have demonstrated that VEGF and its receptors are developmentally regulated and operational in the endothelial cells of the ductus [67]. Thus, VEGF encourages the in growth of vasa vasorum and the recruitment of mononuclear cells to the intima of the ductus, which represents a crucial stage in the anatomical restructuring of the ductus arteriosus [68,69]. Based on this hypothesis, Sallmon et al. [70] investigated whether the VEGF polymorphism rs2010963 status is associated with the incidence of patent arteriosus ductus and/or the efficiency of pharmacological treatment. The authors assessed the rs2010963 status in a cohort study that included 814 preterm infants with a weight less than 1500 g at birth using restriction fragment length polymorphism analysis. Their findings indicate that VEGF rs2010963 polymorphism status is not significantly associated with the rate of patent arteriosus ductus and is not correlated with the success of cyclooxygenase inhibitor treatment [70].

Despite the growing body of evidence linking VEGF polymorphisms to CHD, more research is needed to fully understand the mechanisms by which these polymorphisms contribute to CHD risk. Future studies may also explore the potential for using VEGF polymorphisms as biomarkers for CHD risk assessment and as targets for therapeutic interventions to prevent or treat CHD.

### 3.3. Lung Development and Pathologies

Normal lung function and development are crucial for the transition of newborns to the extrauterine environment. The lung tissue expresses VEGF at significant levels due to their high vasculature content, which is critical for gas exchange [71]. Thus, optimal expression of VEGF is necessary for the proper development of the lungs, as it contributes to the development of vascular structures during embryonic development. VEGF acts as a mitogen factor, and a differentiation factor for various lung cells, including endothelial cells and type II pneumocytes [72,73]. VEGF affects not only the vascular side but also the alveolar septum wall, as transient inactivation of the VEGF lung gene results in cellular apoptosis in the alveolar septum wall, airspace enlargement, and increased lung compliance. The essential regulatory roles of epithelial-expressed VEGF in the lung for function and development have been confirmed by animal models.

Moreover, VEGF promotes the growth and differentiation of the type II cells responsible for producing surfactant, which is critical for normal lung function. Decreased levels of VEGF have been correlated to abnormal capillary endothelial cells and bronchopulmonary dysplasia (BPD) [72,74], defined as the need for supplemental oxygen for at least 28 days or oxygen dependency at 36 weeks postmenstrual age. BPD affects from one-fifth to one-third of infants born before 30 weeks of gestation and with a birth weight below 1000 g and is associated with high morbidity and poor neurodevelopmental outcomes [75,76]. The pathogenesis of BPD is multifactorial and involves the developmental arrest of the premature lung, oxygen toxicity, infection, and ventilation-induced lung injury [77].

In some preterm infants, assisted ventilation is required to initiate breathing, but at the same time, it can induce lung injury due to high pressures, high volumes, and oxygen exposure. The inflammatory pathways can be activated by these factors and perpetuate the inflammatory mechanisms triggered by mechanical ventilation, leading to distorted alveolar and vascular growth and alterations in the structure and differentiation of mesenchymal cells in the newborns’ lungs. Hyperoxia from oxygen administration, as well as hypoxemic events, increase reactive oxygen species (ROS) production and tissue inflammation, further contributing to the development of BPD [22,78], which suggests that, besides prematurity and mechanical ventilation, which are the significant risk factors for BPD, there is also individual susceptibility to BPD, and emerging evidence shows that BPD occurs secondary to genetic-environmental interactions in an immature lung [79,80].

In the human population, the VEGF gene is highly polymorphic, and studies suggest it may contribute to the genetic predisposition of diseases in which angiogenesis plays a role, including BPD. Although studies have not found a direct connection between specific genetic variants and BPD, various genes, pathways, and variants associated with BPD susceptibility have been identified, suggesting that each genetic variant may play a small part in BPD development. Together, they significantly alter the normal lung development of preterm infants [81].

The +936C/T (rs3025039) genotype is among the most widely studied polymorphisms for association with different pathologies [82,83,84]. Kwinta et al. [85] found that VEGF-460T>C polymorphism may influence the risk of BPD in a study conducted on 181 newborns with a mean gestational age of 28 weeks. Filonzi et al. [86] investigated the potential relationship between the +936C/T VEGF and -710C/T VEGFR1 polymorphisms and the incidence of BPD. The study included 82 very low birth-weight newborns, of which 33 infants developed BPD and 49 served as controls. Significant statistical differences were observed between the newborns with BPD and the controls regarding factors, such as maternal preeclampsia, chorioamnionitis, birth weight, gestational age, mechanical ventilation, and duration of oxygen therapy. However, no differences were observed at the genotypic or allelic levels with respect to the VEGFR1 and VEGF molecular polymorphisms. Therefore, no association was established between these genetic variants and BPD.

Poggi et al. [43] conducted a retrospective study to examine the impact of particular gene polymorphisms encoding VEGF-A (rs1547651, rs833058, rs833061, and rs3025039) on the incidence of BPD in a group of premature infants. The authors established a correlation between the presence of these polymorphisms and the development of BPD. According to their findings, VEGF-A might have an independent effect on birth weight and gestational age and could function as a protective or risk marker for complications associated with prematurity, including BPD.

Fujioka et al. [87] demonstrated a significant difference in the distribution of the -634C>G allele and genotype among premature Japanese newborns with BPD. They also identified a haploblock that contained the -634C>G allele, and a significant reduction in the frequency of the -634C/-7C haplotype, which only includes C alleles, in the BPD population compared to the non-BPD population. The authors hypothesize that premature Japanese babies with the VEGF -634G allele may have lower levels of VEGF in lungs and may be more susceptible to BPD. However, the results of Fujioka et al. cannot be compared directly with these studies, due to the differences in ethnic background, given the fact that ethnicity has a marked influence over the distribution of gene polymorphisms. This represents a limitation of this study, which was focused on a specific homogenous sample of Japanese infants treated in a single hospital.

In the pediatric population, the polymorphism for VEGFA-rs2146323AA was associated with a significantly reduced risk (*p* = 0.03) for wheezing [88] and in asthma development [89].

The role of angiogenic factors, particularly VEGF, is central to vasculogenesis. This importance is substantiated by animal experiments and human studies, which have demonstrated low VEGF levels in plasma and bronchoalveolar lavage in BPD cases. Hence, it is reasonable to speculate that certain functional gene polymorphisms of angiogenic factors might affect the risk of developing BPD. Nonetheless, it is important to note that any association between genotype and disease does not imply a causative relationship, and, therefore, additional prospective studies are necessary to determine the possible role of genetic polymorphisms in BPD and perinatal complications.

### 3.4. Eye Conditions

A significant late-onset neonatal pathology is retinopathy of prematurity (ROP), a vascular proliferative process affecting premature newborns with less than 34 weeks of gestational age, and representing a leading cause of childhood blindness. There are five stages of ROP indicating disease severity from stage 1 (demarcation line) to stage 5 (total retinal detachment). A sign of gravity is also the plus sign, which represents particularly dilated and tortuous blood vessels at the posterior eye pole. [90].

Retinal vascularization begins at about 12 weeks in humans and is completed by 36 to 40 weeks of gestation [91].

In utero, a physiologic state of hypoxia and the presence of a VEGF-mediated environment drives normal vascularization. The predisposition of preterm infants to developing ROP relates to their immature retinal vasculature. In the retina of premature babies, vascularization is achieved through angiogenesis and the proliferation of endothelial cells. This proliferation occurs under the influence of VEGF, which is critical for retinal vascular development [92,93], and the genetic expression of VEGF has an essential role. Retinal hypoxia is associated with a marked increase in VEGF expression. The most important predictors of ROP are prematurity, low birth-weight, intrauterine growth restriction, Apgar scores [94], and possibly genetic polymorphism. Vascular proliferation is associated with abnormal retinal vascular development observed in extremely premature neonates (less than 28 weeks) [95]. In rat models, using anti-VEGF antibodies or VEGF receptor, two inhibitors increase retinal vessel tortuosity and dilation [26].

The studies suggests that some genetic factors influence the phenotypic variability of ROP. The analysis of VEGF 936 C/T and VEGF 634 C/G polymorphisms in a group of 106 premature infants showed that the frequency of the genotype of VEGF 634 CG was significantly higher in those with ROP and that the VEGF 634 C/G polymorphism influences the risk of ROP in infants [96]. In the Egyptian population, the genotype of VEGF 634 GG was significantly associated with the development of plus disease or aggressive posterior ROP (*p* = 0.001), and authors suggest that it may be associated with the severity of the disease rather than disease development [96].

In a Turkish population, there was no correlation between the carrier states of promoter polymorphisms VEGF (-634)C, and the VEGF (-460)C gene and spontaneous regression or progression of ROP in preterm populations [97]. A cohort study of 61 patients with advanced ROP presented no significant difference in the frequency of the allel of the VEGF gene compared with normal subjects and preterms with advanced ROP [98]. The study of Vannay et al. [99] on a Hungarian population of 86 preterm newborns with a birth weight under 1500 g (VLBW) and ROP showed that the VEGF -460TT/+405CC haplotype was more dominant in preterm with treatment than in the preterm without treatment (13 of 86 versus 1 of 115; *p* < 0.001), and the correlation remained significant (*p* < 0.01) even after the correction to risk factors of ROP (gestational age, gender, oxygen therapy). These suggest that the genotype of VEGF may be associated with a risk of evolutive ROP in VLBW infants and a risk for ROP requiring cryotherapy/photocoagulation [99]. These findings suggest that the testing of these VEGF SNPs would provide valuable information for the risk assessment of ROP [99].

Analyzing the data reported [100] on gender, in male preterm infants with severe ROP (ROP stage: 4–5) the prevalence of the VEGF -2578A allele was lower than in male preterm infants without or with mild ROP (ROP stage: 1–3) (*p* = 0.044, adjusted odds ratio [range]: 0.26 [0.07–0.96]).

A case–control study of 65 preterm infants showed that VEGF165 and VEGF121, the two most prevalent human VEGF isoforms, are present in premature infants who did not develop retinopathy [101]. Kwinta et al., in a study on 181 Polish preterm newborns, showed that the carriage of polymorphic allele -460 T was significantly higher in ROP infants who needed treatment as compared to the no ROP infants [102].

In a prospective study from a tertiary center that enrolled 204 Japanese infants (<35 weeks of gestational age), the VEGF (g.+13553C>T, g.-634G>C) polymorphism did not differ between ROP and non-ROP patients [103]. For the Japanese population, a genotype of the VEGF pathway weakly affects the gravity of ROP compared with other clinical factors, such as low birth weight, respiratory distress syndrome, and blood transfusion [103].

A report of a twin pair, monochorionic diamniotic, born at 29 weeks of gestation in which both twins developed severe ROP with retinal detachment suggests that the synergistic effects of unstable cardiopulmonary status and genetic predisposition due to VEGF 936C>T polymorphism caused the development of severe ROP [104].

Stage 3 ROP usually develops in extremely preterm neonates. An unusual case of stage 3 ROP is described by Mandal et al. in an infant born at 30 weeks of gestation and with a birth weight of 2102 g, in whom they found an unbalanced translocation 18p (monosomy), 6p (trisomy), and over-expression of VEGF [105].

A study on 54 preterm infants with ROP of the China Han ethnic population found that SNP for VEGF -165C/T is associated mainly with severe ROP [106].

In a Turkish population of 148 newborns with moderate and severe ROP, it was observed that the overload of mutant alleles in VEGFA rs3025039 and rs2010963 increased ROP severity and treatment requirements (*p* < 0.001, *p* < 0.001) [107].

Luo Y et al. [19] evaluated, in a meta-analysis, in the VEGF gene, four common genetic polymorphisms, VEGF -460T/C, -634G/C, -2578C/A, and +936C/T, and their relationship with the risk of ROP. Statistical analysis showed a significant relationship between VEGF -460T/C and ROP risk, but no relation was identified for VEGF 936C/T, -634G/C, and -2578C/A. One of the most significantly studied VEGF polymorphisms is VEGF 460T/C in the promoter region [19].

The critical role of VEGF in inducing retinal neovascularization prompted researchers to investigate the role of anti-VEGF drugs in the management of ROP [108]. Studies in mice showed defective vascular development and early embryonic lethality when inactivating a single allele of the VEGF gene. In animal models, blockage of the VEGF signaling pathway prevented retinal ischemia-associated neovascularization [109].

In addition to the genetic polymorphism of VEGF, other genetic polymorphisms also implicated the association of candidate genes from various signal pathways in the development of ROP. Because the VEGF gene is situated near the major histocompatibility complex (MHC) region on chromosome 6, the genes of some proteins found here are probably implicated in retinopathy.

In addition to the genetic predisposition, several factors are involved in developing ROP, such as low birth weight, low gestational age, and prolonged oxygen therapy. In some cases, the genetic component determines the risk of progression of ROP to severe forms of the disease. However, a genetic component significantly impacting ROP has yet to be discovered [23].

Fevereiro-Martins et al. [23] review analyze current research involving inflammation and genetic factors in ROP’s pathogenesis. They conclude that perinatal inflammation, infection, and genetic factors contribute to ROP pathogenesis and new technologies involving genomics, bioinformatics, and proteomics may contribute to find genes or pathways associated with ROP and help in the future to discover better solutions in the management of ROP [23].

In a review, Cavallaro et al. [25] conclude that gene mutations and SNPs of ROP-related factors might play significant roles in developing this multifactorial disease. It also launches an idea of how topical propranolol causes anti-VEGF activity by blocking the circulating excess VEGF without affecting the normal vascularization of other organs and systems, and maybe the drug of choice to prevent ROP progression.

In summary, VEGF is recognized to be involved in the developing ROP. For normal retinal angiogenesis, the presence of VEGF is necessary. Still, because of known clinical factors involved in the pathogenesis of ROP, the presence of genetic polymorphisms can be a supplementary risk to the development and severity of ROP.

### 3.5. Hemangioma and Cerebral Pathologies

The most frequent vascular tumor in childhood is infantile hemangioma (IHs) [110]. IHs increase to maximum size when the infants are approximately 9–12 months; after a short plateau phase, IHs progressively regress after one year of age, and most may be in total remission by four years of age. They predominantly involute without significant residua, and the majority do not require treatment. Some IHs have associated structural anomalies. Complications such as ulceration, prevention of disfigurement, and impairment of function or vital structures are indications for intervention.

IHs pathogenesis are multifactorial, and although researchers have proposed various hypotheses, it is still unclear.

Three important mechanisms are involved in IHs pathogenesis. The first includes the placental origin and the metastatic niche [111]. Ref. [112] suggests that molecular markers of hemangioma vessels, such as merosin, laminin, Lewis Y antigen, Fc II gamma receptor, and erythrocyte-type glucose transporter-1 (GLUT-1), are expressed by normal fetal placental microvessels. The second mechanism involves perinatal hypoxia and the renin-angiotensin system [113]. Intrauterine hypoxic stress may determine overexpression of angiogenic factors, such ashypoxia-inducible factor-1 alpha (HIF-1α), which can increase the transcription of downstream target genes such as GLUT-1 and VEGF [114]. The third mechanism involved is vasculogenesis and angiogenesis by stimulating stem cells to differentiate into endothelial cells to create blood vessels under hypoxic conditions [86]. Bone marrow-derived endothelial progenitor cells can be stimulated by tissue ischemia leading to local vasculogenesis [115].

During the first year of life, the rapid proliferation of endothelial cells distinguishes, followed by spontaneous regression, VEGF as one of the potential factors responsible for IHs development. It is a hypothesis that the high rennin serum levels and the local expression of ACE lead to a high concentration of angiotensin II (ATII), which, along with VEGF, drives cell proliferation [116].

A Polish case–control study of 99 children hospitalized with IHs compared with healthy control subjects showed the effect of selected polymorphisms in the genes coding for VEGF-A (+405 G/C, rs2010963; +936 C/T, rs3025039) and its receptor VEGFR-2 (+1416 T/A, rs1870377; -271 G/A, rs7667298) on the susceptibility to infantile hemangioma. Those with at least one G allele of +405G/C VEGF-A polymorphism have a significantly decreased risk of IH [110].

Understanding the pathogenesis of IHs lays the foundation for developing new treatments to reduce adverse reactions.

Intraventricular cerebral hemorrhage (IVH) is a frequently encountered complication in premature newborns. It started in the germinal matrix, which is a fully vascularized collection of precursor cells in the developing brain. The etiology of IVH is multifactorial, and is determined by the fragility of the germinal matrix vasculature and the disturbance in cerebral blood flow [117]. The germinal matrix exhibits rapid endothelial proliferation because the VEGF level and angiopoietin-2 are higher in the germinal matrix than in the cerebral cortex or white matter. VEGF activates rapid angiogenesis in the germinal matrix vasculature. VEGFA can increase the permeability of vessels and cause vasodilatation, and this is associated with various hemorrhagic disorders.

Gong et al. [118], in a study on a Han Chinese population, suggests that the genetic variants of the VEGFA gene may modify cerebral hemorrhage risk in patients with brain arteriovenous malformations (BAVMs). They evaluated whether the rs1547651 variant is associated with cerebral hemorrhage risk in patients with BAVMs. They found one SNP out of nine selected SNPs showed a significant connection with cerebral hemorrhage risk and two haplotypes with a major protective effect. The heterozygous genotype was significantly associated with increased cerebral hemorrhage risk (adjusted OR = 2.11, 95% CI =1.01–4.42) compared with the AA genotype. The results suggest that the VEGFA gene variants might contribute to an increased cerebral hemorrhage risk in patients with BAVMs in a Han Chinese population [118].

A prospective study on 382 infants with a gestational age less than 28 weeks hospitalized in the neonatal intensive care unit analyzed 4 SNPs for VEGF polymorphisms (RS699947, RS2010963, RS3025039, and RS1570360). They established that the GA/AA genotype in VEGF RS1570360 and the AA/AC genotype in VEGF RS699947 were associated with elevated incidence rates of IVH in newborns ≤28 weeks of gestation (*p* = 0.017) [119]. Additionally, a higher presence of the CC genotype for SNP VEGF RS3025039 tended to an increased incidence of IVH of principally African American newborns, and the CC genotype of VEGF RS1570360 was associated with a need for ventriculoperitoneal shunt placement surgery.

The assessment of the VEGF level may be an early predictive factor for the development of IVH and management of its sequelae in preterm neonates. Shimi et al. [120], in a prospective study on 150 preterm neonates less than 34 weeks of gestation, found that the serum level of VEGF increased in IVH compared to the non-IVH group (*p* = 0.001), which suggests that serum VEGF can predict the development of IVH. Additionally, in 10 patients, they sampled cerebrospinal fluid, and the VEGF level was increased, which could indicate the need for permanent shunt placement. Yang et al. [121] used a tetracycline-regulated transgenic system to test whether the induction of VEGF in the germinal matrix leads to intracranial hemorrhage. This genetic approach initially induced a dense system of loosely adjacent endothelial cells and pericytes near lateral ventricles, similar to the immature vascular system in human fetal brains. Biochemical analyses and gene expression prove the role of VEGF in perinatal cerebral hemorrhage and involve its downstream proteases as possible therapeutic targets.

Intraventricular hemorrhage remains an important problem in modern neonatal care because the survival rate for the smallest premature infants continues to increase.

### 3.6. Digestive Pathologies

Another dreaded pathology in the neonatal period is necrotizing enterocolitis (NEC), a complication with an increased risk of morbidity and mortality, especially in extremely premature neonates. The etiopathogenesis is complex and involves multiple factors, including intestinal ischemia after fetal distress [122], intestinal hypoperfusion, infection causing an inflammatory process in the mucosa, and formula feeding. The role of genetics in the pathogenesis of NEC is suggested by epidemiological data that showed a different susceptibility in groups of populations and twins. Among the genetic factors, the polymorphism for VEGF is incriminated. The results suggest that G+405C VEGF polymorphism might be associated with an elevated risk of preterm birth and that C-2578A VEGF polymorphism may contribute to the development of perinatal complications, such as NEC [44]. Studies reveal that decreased intestinal expression of VEGF is seen in human NEC and decreased VEGF signaling increases susceptibility to NEC in a mouse model [123]. On the other hand, a cohort study (358 preterm newborn) could not confirm the association between the VEGF C-2578-A and NEC [124]

Liu et al. [125] studied the link between VEGF polymorphism and biliary atresia in the Southern Chinese Han population and reported no correlation with VEGFA gene +936T/C SNP (rs3025039). Instead, in the Taiwanese population, the VEGF +936C/T polymorphism and particularly the C allele have been associated with biliary atresia, maybe conferring increased susceptibility to the disease [126].

## 4. Limitations and Further Studies

The main limitation of our study is searching in a single database and the small number of articles evaluating the link between VEGF polymorphism and pathology in the neonatal period. Mainly studies have limitations and do not replicate results. Only for the ROP, data suggest that genetic factors play a role in the phenotypic variability of ROP.

Additionally, the variability of the results from one population to another does not allow the generalization of the results.

An essential element that can influence the effects of VEGF polymorphism on perinatal morbidities is the diversity of population and ethnicity (Table 1). This element was researched and demonstrated in the case of ROP. A reasonable basis for a difference among racial groups in susceptibility to ROP might be ocular pigmentation [26].

SNPs (single nucleotide polymorphisms); VEGF (Vascular endothelial growth factor).

Regarding the type of population analyzed, the majority are premature newborns who developed various morbidities and whose data were compared with the same population but without pathology.

Another essential element that must be considered when analyzing the cases in which the genetic polymorphism for VEGF was determined is that the analysis is not unitary. Thus, different sequences are evaluated (Table 1), knowing that VEGF has at least 30 functional single-nucleotide polymorphisms (SNPs).

Studies suggest that VEGF polymorphism is involved in neonatal pathology directly; this aspect is indicated by the results obtained from the ROP analysis or indirectly through the determining role in premature birth and or with growth restriction indirectly influencing perinatal morbidity (Figure 1).

## 5. Conclusions

VEGF polymorphisms play a critical role in placental and fetal development, thus determining neonatal pathology.

They are associated with early- (cerebral hemorrhage, hemangioma, necrotizing enterocolitis) and late-onset neonatal pathology (retinopathy of prematurity, bronchopulmonary dysplasia).

The involvement of VEGF and VEGF polymorphism has been demonstrated for retinopathy of prematurity. Identifying genetic influences may help improve the screening programs for neonatal morbidity and special late morbidity (ROP, BPD).

There is evidence linking VEGF polymorphisms to congenital heart defects and long-term survival after surgical correction. However, further research is needed to understand better the mechanisms by which VEGF polymorphisms play a role in the development of cardiovascular diseases and to identify possible therapeutic targets to treat these conditions.

Functional gene polymorphisms of angiogenic factors might affect the risk of developing bronchopulmonary dysplasia, a disease with multifactorial pathogenesis, in which the developmental arrest of immature lungs is enmeshed with other factors such as infection, the toxicity of oxygen-reactive species, and injury induced by mechanical ventilation.

Necrotizing enterocolitis is a complex etiopathogenesis involving gut hypoperfusion and ischemia, inflammation of the intestinal mucosa, and formula feeding. Alongside these well-studied factors, it appears that decreased intestinal expression of VEGF could increase susceptibility to necrotizing enterocolitis. Further studies investigating this genetic susceptibility may improve the management of neonates with known risk factors and have a positive impact on the long-term outcome. Gene expression and biochemical analyses support the causative role of VEGF in perinatal cerebral hemorrhage, which represents a significant complication causing potential long-term disabilities. Due to advances in neonatal intensive care, the survival rate of extremely preterm neonates, with the highest risk for developing IVH, has increased. Determining the VEGF level in these immature infants may be useful as an early predictive factor for the development of cerebral hemorrhage and the management of long-term sequelae. In conclusion, the link between VEGF polymorphisms and neonatal pathologies represents a territory that is worth analyzing for developing new treatments with reduced adverse reactions.

## Figures and Tables

**Figure 1 children-10-00744-f001:**
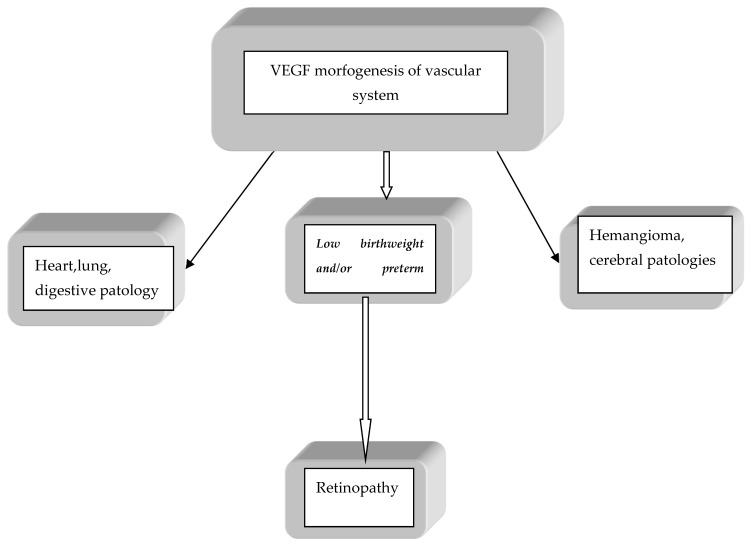
Link between VEGF polymorphism and neonatal pathology.

**Table 1 children-10-00744-t001:** Type of polymorphism.

Country Origin of the Studied Population	Type of Study	Inclusion Number	Type of Polymorphism	Reference
			Pregnancy	
Australia	Cohort	1169	VEGF 634CC	Andraweera PH et al. (2012) [40]
Romania	Case-control	164	VEGF-CT936	Procopciuc LM et al. (2014) [15]
Ecuador	Case-control	31	VEGF: -2578A/C, -1498C/T, -1154A/G, -634C/G and +936C/T.	Chedraui P et al. (2013) [39]
Poland	Cohort	67	VEGF SNP	Iciek R et al. (2014) [38]
Greece	Case-control	133	VEGF 936C/T	Galazios G et al. (2009) [17]
Turkey	Case-control	123	VEGF +813	Atis A et al. (2012) [18]
Italia	Cohort	342	*VEGFA* rs1547651, rs833058, s833061, rs3025039	Poggi C et al. (2015) [43]
			Heart pathologies	
USA	Cohort	422	VEGFA rs833069 minor allele	Mavroudis CD et al. (2018) [63]
Italia	Case-control	198	VEGFA rs3025039 SNPs	Balistreri CR et al. (2020) [27]
Chile	Case-control	122	VEGFA c.-2578C>A (rs699947), c.-1154G>A (rs1570360), c.-634C>G (rs2010963)	Calderón JF et al. (2009) [65]
USA	Cohort	550	VEGFA rs833069	Kim DS et al. (2014) [62]
Pakistan	Case-control	242	VEGF rs699947 (c.-2578C > A)	Sarwar S et al. (2021) [59]
Pakistan	Case-control	350	VEGF, rs36208048 (NG_008732.1:g.3877C > A	Sarwar S et al. (2022) [60]
			Lung diseases	
Finland, Canada	Case-control	160	six tagging single nucleotide polymorphism (tSNPs)	Mahlman M et al. (2015) [80]
Japan	Case-control	97	VEGF -1498T>C, -1154G>A, -634C>G, -7C>T, 936C>T, and 1612G>A	Fujioka K et al. (2014) [87]
Germany	Case-control	155	VEGF rs699947, rs2010963, rs3025039	Mailaparambil B et al. (2010) [75]
Italy	Case-control	82	VEGF+936 C/T	Filonzi L et al. (2022) [86]
Poland	Case-control	181	VEGF -460T>C and 405G>C	Kwinta P et al. (2008) [85]
Italy	Case-control	238	VEGFA-rs833058CT, VEGFA-rs2146323AA	Esposito S et al. (2014) [88]
Denmark	Cohort	411	13 SNPs.	Kreiner-Møller E et al. (2013) [89]
			Eye conditions	
Japan	Case-control	204	VEGF(g.-634G>C, +13553C>T)	Kusuda T et al. (2011) [103]
United Kingdom	Case-control	188	VEGF -634 G>C, VEGF 936C>T	Cooke RW et al. (2004) [93]
United Kingdom	Study case	1	Chromosomal VEGF analysis	Mandal K et al. (2007) [105]
Turkey	Case-control	148	VEGFA rs2010963 and rs3025039	Ilguy S et al. (2021) [107]
Hungary,	Case-control	200	VEGF(-2578) A and G”allele polymorphisms	Bányász I et al. (2006) [100]
USA	Case-control	122	VEGF promoter region (containing -634 G>C and -460C>T polymorphism)	Shastry BS et al. (2007) [98]
China	Case-control	174	VEGF -165C/T, -141A/C, T-165C-141 and C-165A-141 haplotypes	Zang S et al. (2017) [106]
Egypt	Case-control	102	VEGF 634 C/G and 936 C/T polymorphisms	Ali AA et al. (2015) [96]
Turkey	Case-control	123	VEGF (-634) C and (-460) C polymorphisms	Kaya M et al. (2013) [97]
Iran	Case-control	111	VEGF +405 (rs2010963) and VEGF +936 (rs3025039)	Kalmeh ZA et al. (2013) [92]
Hungary and Poland	Case-control	181, 211	VEGF 405G>C and -460T>C	Vannay A et al. (2005) [99] and Kwinta P et al. (2008) [102]
Japan	Case report	2	VEGF 936C>T polymorphism	Fujioka K et al. (2013) [104]
Hungary	Case-control	237	VEGF T(-460)C, G(+405)C, and C(-2578)A	Dunai G et al. (2008) [94]
			Hemangioma and cerebral pathologies	
USA	Case-control	382	VEGF (RS699947, RS2010963, RS3025039, and RS1570360)	Prasun P et al. (2018) [119]
Poland	Case-control	99	VEGF-A (+405G/C, rs2010963; +936 C/T, rs3025039	Oszajca K et al. (2018) [110]
			Digestive pathologies	
Netherlands, Spain, Italy	Case-control	358	VEGF C-2578A (rs699947)	Moonen RM et al. (2020) [124]
Hungary	Case-control	328	VEGF T-460C, C-2578A, G+405C	Bányász I et al. (2006) [44]
China, Han population	Case-control	1979	VEGFA rs3025039	Liu F et al. (2018) [125]
Taiwan	Case-control	205	VEGF (-2578 A/C, -634 G/C, and +936 C/T)	Lee HC et al. (2010) [126]

## Data Availability

Not applicable.

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
