# Peer review of "Genetic Polymorphisms of Vascular Endothelial Growth Factor in Neonatal Pathologies: A Systematic Search and Narrative Synthesis of the Literature"

_children, 2023, doi:10.3390/children10040744_

Round 1

Reviewer 1 Report

Congratulations on the extensive work!

The paper is very interesting and it helps understand the mechanisms behind some neonatal pathologies. It is obvious that the authors gathered a lot of data regarding this subject. I would recommend extending the conclusion paragraph.

There are some errors using the references inside the text. Please correct them: 

- Keshavarzi et al. [Error! Bookmark not defined.]

- Procopciuc et al. [Error! Bookmark not defined.]

- Atis et al. [Error! Bookmark not defined.]

- bronchopulmonary dysplasia (BPD) [72, 74Error! Book- 339 mark not defined.]

- the development of BPD [78, Error! Bookmark not 353 defined.]

Author Response

Thank you for your appreciation and suggestions. We have corrected the errors related to the references and performed a spell check to correct the English language mistakes. Also, we have expanded the conclusions of the study by adding a few paragraphs.

Thank you.

Hasmasanu Monica

Reviewer 2 Report

Thank you for the opportunity to review this interesting manuscript reviewing the available evidence concerning the potential role of VEGF polymorphisms on neonatal morbidity. Your review of the current literature suggests a possible association, but you have thoroughly addressed the limitations of the review, including the fact that only scant evidence is currently available. This was a pleasure to read and certainly thought provoking. 

Author Response

Thank you for your words of appreciation. We have conducted a spell check and corrected the English language mistakes.

Thank you

Dr Hasmasanu Monica

Reviewer 3 Report

This review covers the evaluated the available literature about the association between the genetic polymorphisms of VEGF and neonatal pathology. It is a well written manuscript. I would suggest the authors to add a flowchart of study selection during the literature search. In addition I would suggest the authors to shorten the text.

Author Response

Thank you for your appreciation and comments. I didn't add the flowchart with the methodology because I already explained it in the text and I didn't want to make the manuscript too long. We have conducted a spell check and corrected the English language mistakes.

Thank you,

Dr Hasmasanu Monica